# IL-33 and MRGPRX2-Triggered Activation of Human Skin Mast Cells—Elimination of Receptor Expression on Chronic Exposure, but Reinforced Degranulation on Acute Priming

**DOI:** 10.3390/cells8040341

**Published:** 2019-04-11

**Authors:** Zhao Wang, Sven Guhl, Kristin Franke, Metin Artuc, Torsten Zuberbier, Magda Babina

**Affiliations:** Department of Dermatology, Venerology and Allergy, Charité-Universitätsmedizin Berlin, Corporate Member of Freie Universität Berlin, Humboldt-Universität zu Berlin, and Berlin Institute of Health, Charitéplatz 1, 10117 Berlin, Germany; wang.zhao@charite.de (Z.W.); sven.guhl@gmx.de (S.G.); kristin.franke@charite.de (K.F.); metin.artuc@charite.de (M.A.); torsten.zuberbier@charite.de (T.Z.)

**Keywords:** mast cells, skin, IL-33, pseudo-allergy, neurogenic mast cell activation, neuropeptides, MRGPRX2, degranulation, JNK, p38

## Abstract

Clinically relevant exocytosis of mast cell (MC) mediators can be triggered by high-affinity IgE receptor (FcεRI)-aggregation (allergic route) or by the so-called pseudo-allergic pathway elicited via MAS-related G protein-coupled receptor-X2 (MRGPRX2). The latter is activated by drugs and endogenous neuropeptides. We recently reported that FcεRI-triggered degranulation is attenuated when human skin mast cells are chronically exposed to IL-33. Here, we were interested in the regulation of the MRGPRX2-route. Chronic exposure of skin MCs to IL-33 basically eliminated the pseudo-allergic/neurogenic route as a result of massive MRGPRX2 reduction. This downregulation seemed to partially require c-Jun N-terminal Kinase (JNK), but not p38, the two kinases activated by IL-33 in skin MCs. Surprisingly, however, JNK had a positive effect on MRGPRX2 expression in the absence of IL-33. This was evidenced by Accell^®^-mediated JNK knockdown and JNK inhibition. In stark contrast to the dampening effect upon prolonged exposure, IL-33 was able to prime for increased degranulation by MRGPRX2 ligands when administered directly before stimulation. This supportive effect depended on p38, but not on JNK activity. Our data reinforce the concept that exposure length dictates whether IL-33 will enhance or attenuate secretion. IL-33 is, thus, the first factor to acutely enhance MRGPRX2-triggered degranulation. Finally, we reveal that p38, rarely associated with MC degranulation, can positively affect exocytosis in a context-dependent manner.

## 1. Introduction

MAS-related G protein-coupled receptor-X2 (MRGPRX2) has become a recent focus in mast cell (MC) research. The G-protein coupled receptor triggers so-called pseudo-allergic and neurogenic reactions by degranulating MCs independently of the classical FcεRI-mediated allergic route, but the ensuing symptoms clinically phenocopy those arising from allergic MC stimulation [1,2,3,4]. The discovery of MRGPRX2 and its murine equivalent Mrgprb2 marks a paradigm change in MC biology [5] and can now explain observations that remained inexplicable in the past, including the divergent response patterns among MCs from different tissues to basic secretagogues and neuropeptides [6,7].

In fact, depending on the content of secretory granules, human MCs are categorized into two subtypes [8]. Most lung and gut MCs belong to the so-called MC_T_ category and express only tryptase, while most connective tissue MCs are of the MC_TC_ type, characterized by the presence of tryptase and chymase, and some other characteristics [9,10]. MC_T_ roughly correspond to MMC (mucosal MCs) in the mouse, whereas the murine correlate of MC_TC_ is the CTMC subset (connective tissue type MCs) [1,5].

Interestingly, MRGPRX2 (or its murine equivalent Mrgprb2) is highly expressed in the MC_TC_/CTMC subsets, whereas MC_T_/MMC-type MCs lack the receptor [5,11,12,13]. This expression pattern now makes clear why the latter types are refractory to secretagogues like compound 48/80 and Substance P (SP), which have been used to activate human skin MCs (nearly all MC_TC_) and rodent CTMC for decades without knowing the receptor and mechanism involved [6,7].

The clinical importance of MRGPRX2 is underlined by the plethora of endogenous and exogenous ligands, which trigger its activation. They comprise antimicrobial host defense peptides (e.g., cathelicidin), neuropeptides (including SP and somatostatin), and various FDA-approved cationic drugs known to trigger pseudo-allergic reactions in susceptible individuals [1,11,14,15]. We recently reported that the allergic and pseudo-allergic/neurogenic degranulation routes are independently controlled at the population level [16]. This was in accordance with the two patterns of granule exocytosis, as described by Gaudenzio and colleagues [17]. In their study, MRGPRX2-mediated degranulation was a rapid process, associated with a quick and transient peak of Ca^2+^ influx, followed by secretion of individual granules, whereas FcεRI-elicited secretion was delayed, but progressive, and characterized by granule-to-granule fusion, also termed compound exocytosis [17].

Surprisingly, we also observed that MRGPRX2- and FcεRI-triggered activation routes are inversely regulated by stem cell factor (SCF) [16], the most crucial supportive factor of the MC lineage [18]. In our studies, SCF did not only attenuate pseudo-allergic/neurogenic responses when provided acutely to skin MCs, i.e., directly before stimulation, but it also dampened MRGPRX2 expression and function during skin MC culture, where it co-operated with IL-4 to inhibit the alternative route [19]. Skin MCs exposed to retinoic acid likewise diminished MRGPRX2 function [20]. Conversely, the allergic route was simultaneously promoted by all of the above stimuli, i.e., SCF, IL-4, and retinoic acid [19,20].

IL-33 is another cytokine crucially implicated in MC biology [21,22,23]. We recently reported that chronic exposure of human skin MCs to IL-33 dampens their ability to respond to FcεRI aggregation [24]. Because others had reported that IL-33 boosted allergic degranulation when given acutely [25,26,27,28] but led to lowered secretion after long-term exposure [29], the combined data indicated a dichotomy between short and persistent contact with the cytokine.

In the current study, we examined the role of IL-33 in the context of MRGPRX2-triggered activation of skin MCs. We report that long-term contact with IL-33 virtually eliminates expression of MRGPRX2 and therefore erases MC responsiveness to MRGPRX2 ligands. In stark contrast, a short burst of IL-33 primes the pseudo-allergic/neurogenic (and also the allergic) route, and this enhancement occurs in a p38-dependent manner. The current data further emphasize the double-edged character of IL-33 in skin MC biology. They also reveal IL-33 as the first factor described to date to positively influence the MRGPRX2 cascade in skin MCs and to act congruently on the allergic and pseudo-allergic/neurogenic routes.

## 2. Materials and Methods

### 2.1. Isolation and Culture of Human Skin MCs

MCs were obtained from human foreskin and breast skin tissue by a routinely employed technique [13,20,30,31,32,33]. Human skin tissue was obtained from circumcisions or cosmetic breast reduction surgeries, with informed consent provided by the patients or their legal guardians and approval by the university ethics committee. The experiments were performed according to the Declaration of Helsinki Principles. Breast skin samples were from single donors, while foreskins were pooled from several donors (typically 2–9) and used as one preparation to obtain the required cell numbers.

The skin was cut into strips and treated at 4 °C overnight with 0.5 mg/mL dispase (BD Biosciences, Heidelberg, Germany). The epidermis was removed, and the dermis was smashed and digested with 150 mg/mL collagenase (Worthington, Lakewood, NJ, USA), 75 mg/mL hyaluronidase (Sigma, Deisenhofen, Germany), and DNase I at 10 μg/mL (Roche, Basel, Switzerland) at 37 °C in a shaking water bath for 1 h. The cells were filtered from remaining tissue. MC purification was achieved with anti-human c-Kit microbeads and the Auto-MACS separation device (both from Miltenyi-Biotec, Bergisch Gladbach, Germany). MC purity, as assessed by acidic toluidine-blue staining, was >98%. Viability by trypan blue exclusion was >99%.

### 2.2. MC Treatment

For long-term culture experiments, mostly breast skin-derived MCs were kept in SCF alone (100 ng/mL) or SCF and IL-33 (20 ng/mL) for 5 weeks, exactly as described [24]. Cytokines were provided once a week and cultures were automatically counted (by CASY-TTC, Innovatis/Casy Technology, Reutlingen, Germany) [30,31,32,33] and re-adjusted to 5 × 10^5^/mL on a weekly basis. Two preparations of foreskin MCs were used for 5-week cultures analogously to breast skin MCs (Appendix A).

For the other experiments, mainly foreskin MCs were used upon reaching optimal proliferation with SCF (100 ng/mL) and IL-4 (20 ng/mL) or SCF alone [13,33,34]. Cells were deprived of cytokines for 16 h prior to downstream experiments. For priming experiments, cells were pretreated with no additive or IL-33 (20 ng/mL) for 30 min. For kinase inhibition, cells were pre-incubated with SP600125 (JNK inhibitor, 5 µM) or SB203580 (p38 inhibitor, 5 µM) for 15 min (both from ApexBio, Houston, TX, USA) prior to the addition of IL-33 (i.e., 45 min prior to the stimulus).

### 2.3. Accell^®^-Mediated RNA Interference

RNA interference in MCs was performed according to an established protocol [24,35,36] using the Accell^®^ siRNA technology (Dharmacon, Lafayette, CO, USA). Briefly, MCs were washed with 1 × Accell siRNA medium (supplemented with Non-Essential Amino Acids and L-Glutamine), plated at 1 × 10^6^/mL in Accell siRNA medium and treated with 1 µM JNK-targeting siRNA (E-003514-00-0010) or non-targeting siRNA (D-001910-10-50, serving as control) for 48 h. After incubation, cells were treated with or without IL-33 for the times indicated.

### 2.4. Flow Cytometry

Flow-cytometry was performed according to routine protocols [30]. Briefly, MCs were blocked for 15 min at 4 °C with human AB-serum (Biotest, Dreieich, Germany) and incubated with specific antibodies for 30 min at 4 °C. The antibodies were as follows: 0.5 µg/mL of anti-human FcεRIα-FITC (clone AER-37, eBioscience, San Diego, CA, USA), Mouse IgG2b, κ-FITC served as isotype control; 0.15 µg/mL PE-labelled anti-human MRGPRX2 (clone K12H4, Biolegend San Diego, CA, USA), mouse IgG2b-PE (clone eBWG2b, eBioscience) served as isotype control. After washing, cells were measured on the Facscalibur (BD Biosciences, San Jose, CA, USA) and analyzed with the FowJo analysis software (FlowJo LLC, Ashland, OR, USA). Net mean fluorescence intensity (MFI), i.e., “MFI specific antibody–MFI isotype control” served for receptor quantification. Negative values were set as 0.

### 2.5. Determination of Histamine Release

Histamine release assays were performed by a method routinely employed in our laboratory [16,30,31,32,33]. In brief, cell suspensions were divided into aliquots, washed twice with PAC-CM (piperazine-*N*,*N*′-bis[2-ethanesulfonic acid]-albumin-glucose buffer containing 3 mM CaCl_2_ and 1.5 mM MgCl_2_, pH 7.4), resuspended, and challenged at 1 × 10^5^/mL by compound 48/80 (10 μg/mL), SP (30 µmol/L), or no stimulus (spontaneous) for 30 min at 37 °C. Supernatants were collected by centrifugation and stored at −20 °C. Complete cellular histamine content was assessed upon cell lysis with 1% perchloric acid. Histamine in the supernatants and complete histamine were measured by an automated fluorescence method (Borgwaldt Technik, Germany, Hamburg). The net histamine release (%) was calculated as [(stimulated release − spontaneous release)/complete histamine present in the MC preparation] × 100.

### 2.6. β-Hexosaminidase Release Assay

Cell suspensions were washed twice and resuspended at 5 × 10^5^ cells/mL in PAC-CM buffer. Aliquots of 100 μL were seeded into 96-well plates and stimulated by FcεRI-aggregation (AER-37 at 0.1 µg/mL, compound 48/80 (10 μg/mL), SP (30 µmol/L), or kept in buffer only. After incubation for 30 min, supernatants (SNs) were collected by centrifugation at 500× *g*, 4 °C for 3 min, and the pelleted MCs rapidly frozen at −80 °C. After thawing, aliquots of 50 μL of 4-methyl umbelliferyl-*N*-acetyl-beta-d-glucosaminide (Sigma-Aldrich, Munich, Germany) solution at 5 μM in citrate buffer (pH 4.5) were mixed with the same volume of supernatant or lysate and incubated for 60 min at 37 °C to measure the level of secreted and cell-remaining β-hexosaminidase. The reaction was stopped by adding 100 mM sodium carbonate buffer (pH 10). Optical density (OD) was measured at 405 nm. Percent β-hexosaminidase release was calculated as: [OD SN/(OD SN + OD lysate)] × 100. Net release was calculated by subtracting spontaneous release, as in the histamine release assay above.

### 2.7. Reverse Transcription-Quantitative PCR (RT-qPCR)

Total RNA was isolated using the RNeasy Total RNA Kit, digested with RNase free DNase (Qiagen, Hilden, Germany), and PCR carried out with the LC Fast Start DNA Master SYBR Green kit (Roche Applied Science). Primers for MRGPRX2 were 5′-GGATCAGGAAGACCGGGATCA and 5′-CGGCCTGGGGAACAGAAAGT. The values were normalized to the housekeeping genes β actin, cyclophilin B, and GAPDH (Glyceraldehyde 3-phosphate dehydrogenase), each ratio contrasted against control conditions (set as 1) and the mean of the three determinations was used for the analysis and is depicted in the figures [30,31,33].

### 2.8. Immunoblot Analysis

Detection of JNK and p38 activation was performed exactly as described [24]. In brief, MCs deprived of growth factors were stimulated with IL-33 (20 ng/mL) for 15 min or kept without stimulus (control), then boiled in Laemmli buffer, and lysates resolved by SDS-PAGE (Sodium Dodecyl Sulfate Polyacrylamide Gel Electrophoresis). After transfer to a blotting membrane and incubation with antibodies, proteins were visualized by a chemiluminescence assay (Weststar Ultra 2.0, Cyanagen, Bologna, Italy) and bands recorded on a chemiluminescence imager (Fusion FX7 Spectra, Vilber Lourmat, Eberhardzell, Germany). The following primary antibodies from Cell Signaling Technology (Frankfurt am Main, Germany) were used: Anti-p-p38 (Thr180/Tyr182, #9211), anti-p38 (#9212), anti-p-JNK (T183/Y185, #9251), and anti-JNK (#9252). Bands were quantified by densitometry with the software ImageJ (National Institutes of Health, Bethesda, MD, USA) and the degree of phosphorylation was assessed by the following equation: Ratio of phosphorylated protein = density_phospo-protein_/density_total protein_.

### 2.9. Statistics

For 2-sample comparisons, differences between groups were assessed by paired *t*-test or by Wilcoxon matched-pairs signed rank test (when not normally distributed) or by one-sample *t*-test (when normalized to control). For more than two groups, Kruskal–Wallis test (non-parametric ANOVA) with Dunn′s multiple comparisons test was employed. Statistical analyses were performed with GraphPad-Prism 7 (San Diego, CA, USA). *p* < 0.05 was considered as statistically significant.

## 3. Results

### 3.1. Skin MCs Lose Responsiveness to MRGPRX2 Ligands and Massively Downregulate MRGPRX2 Expression after Long-Term Exposure to IL-33

Our previous study indicated that chronic exposure to IL-33 reduces FcεRI expression and responsiveness to its aggregation [24]. To reveal an effect of IL-33 on the alternative pseudo-allergic/neurogenic route, MRGPRX2-elicited degranulation was assessed after culture of skin MCs in the presence of IL-33 for 5 weeks. Using breast-skin derived MCs, we found that MRGPRX2-triggered degranulation was severely hampered by IL-33, as evidenced with an exogenous and an endogenous ligand, respectively, i.e., compound 48/80 (C48/80) and SP (Figure 1a). The effect was consistent and found for MCs from every single donor (Figure 1a).

Addressing the reason behind this phenomenon, we detected that IL-33 curtailed MRGPRX2 expression, both at the mRNA (Figure 1b) and protein level (Figure 1c,d). In several MC preparations, MRGPRX2 expression in IL-33-high surroundings was below detection, explaining their resistance to MRGPRX2 ligands (Figure 1a). Foreskin MCs showed the same behavior on long-term culture with IL-33 as breast skin MCs (Appendix A), implying that the effect was universal and independent of sex, age, and precise skin site.

Collectively, long-term IL-33 diminishes MRGPRX2 expression and thereby restrains MC responsiveness to its ligation.

### 3.2. IL-33-Triggered Downregulation of MRGPRX2 Is Partially JNK-Dependent, although JNK Is A Positive Regulator of MRGPRX2 in the Absence of IL-33

We set out to address the mechanism beyond the notable downregulation of MRGPRX2. Because the use of inhibitors for prolonged times (like five weeks) would have been impractical, we first assessed with a time-course analysis after what time MRGPRX2 downregulation commenced following the addition of IL-33. This approach revealed that the decrease at transcript level was rapid (detectable at 2–4 h after the addition of IL-33) but still detectable at 48 h without re-addition of IL-33 (Appendix A). The 4-h time point was selected for further experiments (and based on this, the 24 h point was chosen for the analysis of protein expression). The rapid response to IL-33 made pharmacological interference and knockdown experiments feasible without concerns about indirect effects (likely accumulating over a five-week period and precluding proper interpretation).

We recently reported that among several signaling intermediates, JNK and p38 were the ones activated by IL-33 in skin-derived MCs [24]. Here, we reproduced this pattern by demonstrating JNK and p38 phosphorylation 15 min upon IL-33 administration (Appendix A).

To address the potential involvement of the two kinases in the regulation of MRGPRX2 by IL-33, we employed selective inhibitors and Accell^®^-facilitated knockdown (KD). The latter strategy used a recently established protocol which enables us to perturb endogenous levels of proteins in tissue MCs [24,36].

Surprisingly, we found that the JNK inhibitor alone (but not the p38 inhibitor) dampened MRGPRX2 expression (Figure 2a,b). This was duplicated by RNAi-mediated JNK-KD (Figure 2c,d, for knockdown efficiency see Appendix A). Considering this baseline effect of JNK interference on the expression of MRGPRX2, we wondered whether reduced JNK function and IL-33 would give additive effects when applied to cells together. However, this was not the case, as their combined application did not further reduce MRGPRX2 vis-à-vis each treatment alone. Conversely, inhibition of JNK in the context of IL-33 resulted in higher MRGPRX2 expression (compared to JNK perturbation alone) both at the protein and mRNA level, while the inhibition of p38 had no effect (Figure 3a,b). The same tendency was found for JNK-specific siRNA, i.e., a slight reversal in MRGPRX2 downregulation by IL-33 could be noted, although the extent was lesser compared to the JNK inhibitor (Figure 3). Notwithstanding, with all combinations of strategies and readouts, the ratio “with IL-33/without IL-33” was enhanced after interference with JNK, meaning that JNK perturbation in the presence of IL-33 slightly rescued MRGPRX2 versus the same treatment in its absence. Note that regulation of MRGPRX2 at the transcript level was typically more pronounced than that of the respective protein but that both processes were consistent in their direction.

We conclude that JNK triggered by IL-33 is partially responsible for the attenuated expression of MRGPRX2 in IL-33-rich surroundings. Paradoxically however, in the absence of IL-33, JNK helps maintain MRGPRX2 expression.

### 3.3. Short-Term Priming by IL-33 Fosters Skin MC Degranulation

Having found that IL-33 was a potent negative regulator of MRGPRX2 function through restrained expression, we asked whether IL-33 may influence MC degranulation acutely (i.e., after short preincubation). By applying the endogenous MRGPPRX2 ligand SP or the exogenous secretagogue C48/80 [4,5,37], we found that IL-33 indeed augmented secretion triggered by both ligands (Figure 4a,b). In addition, degranulation elicited by FcεRI aggregation, studied for comparison, was similarly increased (Figure 4d). This was of interest, as MRGPRX2- and FcεRI-elicited mast cell degranulation pathways are unconnected, as reported by us previously [16].

To assess whether the augmented secretory competence was due to rapid changes in receptor expression, we analyzed MRGPRX2 and FcεRI on the cell surface after short-term exposure to IL-33. IL-33 did not change either MRGPRX2 or FcεRI expression (Figure 4c,e), which indicated that increased degranulation did not stem from altered receptor expression but was rather related to signaling.

Together, IL-33 primes for enhanced degranulation by the pseudo-allergic/neurogenic (and allergic) route in skin MCs, and this effect is not caused by altered receptor expression.

### 3.4. Reinforced Degranulation by IL-33 Depends on p38

Considering the unchanged receptor expression, but positive effect of IL-33 on degranulation triggered by both routes, we asked whether the increment was brought about by p38, JNK, or both. To discriminate between the possibilities, we pretreated MCs with p38 or JNK inhibitor prior to the addition of IL-33, and then exposed them to the secretagogues. The p38 inhibitor basically reversed the priming effect of IL-33 in both routes (i.e., MRGPRX2- and FcεRI-mediated), while the JNK inhibitor had no significant effect (Figure 5).

Conversely, neither of the inhibitors influenced degranulation in the absence of IL-33 (Appendix A), indicating that the p38-specific inhibitor exclusively interfered with the increment resulting from IL-33 action.

Together, IL-33 exerts its priming effect on pseudo-allergic/neurogenic and allergic degranulation of skin-derived MCs by activating p38.

## 4. Discussion

Survival and function of MCs in their natural habitat are regulated by an intricate network of growth factors, and cell-cell- as well as cell-matrix interactions, together forming the MC-supportive niche. In this regard, the microenvironment is not only crucial for the regulation of MC differentiation as such, but it also (re)shapes functional programs of MCs after terminal maturation. IL-33 has gained fame as a MC lineage-supportive and -modulating cytokine. It belongs to the IL-1 superfamily, and acts as an alarmin and pro-inflammatory factor when it is released into the extracellular space upon infection or cell injury [38,39,40]. Due to its inflammatory and Th2-skewing potential [41,42], IL-33 is critical in the induction and maintenance of allergic disorders, such as asthma, atopic dermatitis, and allergic rhinitis [43,44,45]. Accordingly, it is found at increased levels in a variety of skin disorders [46,47,48], and may also play a role in tumorigenesis [49].

MCs are among the primary target cells of IL-33 [50], also owing to their abundant expression of the IL-33 receptor ST2 in comparison with a multitude of primary cells, as uncovered by the comprehensive body-wide expression atlas FATOM5 [13]. IL-33 influences MC biology at various levels and modulates their phenotype, synthetic capacity, and mediator release, but this occurs in a MC-subset-dependent manner. For human skin MCs, we recently reported a yin-yang type response, whereby MCs were numerically strengthened and endowed with greater histamine synthetic capacity, while allergic degranulation elicited by FcεRI aggregation and FcεRI expression were attenuated when cells were chronically exposed to the “alarmin” [24].

Whether the newer pseudo-allergic/neurogenic pathway may be modulated by IL-33 is unclear. The receptor MRGPRX2 that is responsible for these reactions is confined to MC_TC_ type MCs [1], which are basically the only MC subset in human skin [51]. We therefore studied the modulation of MRGPRX2-driven activation by IL-33, placing particular emphasis on the distinction between short-term priming and long-term adaptation effects, and also addressed the mechanisms involved.

Ligands targeting MRGPRX2 are numerous and comprise C48/80, drugs like neuromuscular blocking agents, fluoroquinolones, and icatibant, host defense peptides (e.g., cathelicidin), and neuropeptides like SP [5,11,15,52,53]. Because of the large and still increasing number of secretagogues acting by MRGPRX2 activation, the receptor has become the spotlight of MC research and is linked to a number of diseases triggered by non-immunological MC activation, including injection-site hypersensitivity reactions, chronic urticaria, atopic dermatitis, red man syndrome, periodontitis, and drug anaphylaxis [1,3,5,12,15,53]. In particular, a bidirectional cross-talk between MCs and sensory neurons is widely appreciated, and MCs can be found adjacent to peripheral nerve endings, where they are involved in neurogenic inflammation. By acting as the receptor for a variety of neuropeptides (including Substance P, VIP (Vasoactive intestinal peptide), somatostatin, cortistatin and PACAP (pituitary adenylate cyclase-activating peptide) [1,5,11]), likely MRGPRX2 contributes to neurogenic inflammation. In fact, it was recently demonstrated that MRGPRX2 (and its murine equivalent) mediated mechanical and thermal hyperalgesia, as well as immune cell recruitment by MC cytokine/chemokine release triggered by SP (Substance P) [54]. This is in accordance with another report showing that intradermal injections of PACAP result in cytokine secretion promoting leukocyte recruitment in a MC dependent fashion [55].

In our previous study, FcεRI-mediated degranulation was reduced in MCs from a majority of skin donors and FcεRI expression was simultaneously reduced on long-term IL-33 exposure [24]. To discern IL-33’s role in MRGPRX2-triggered activation in the same setting (mimicking chronic inflammation), the current study first focused on MC responsiveness to MRGPRX2-stimulation upon long-term exposure to IL-33 and concomitant receptor expression. Surprisingly, chronic exposure to IL-33 did not only reduce, but basically eliminated, the pseudo-allergic/neurogenic route in skin MCs, and this was found equally with both MRGPRX2 ligands, namely SP and C48/80. Reduced degranulation was associated with massive downregulation of receptor expression, indicating that dampened stimulability was brought about by a lack of MRGPRX2 at the cell surface. At transcript level, downregulation was already observable a few hours after contact with IL-33, suggesting a fairly rapid mechanism of action.

Knowing that IL-33 activates JNK and p38 in skin-derived MCs [24] (see also Appendix A), we employed pharmacological inhibitors and Accell^®^-mediated knockdown of the kinases as strategies to elucidate the molecular underpinnings behind MRGPRX2 down-regulation. Surprisingly, along the way, we found that JNK seemed a positive regulator of MRGPRX2 expression in its own right. In fact, interference with JNK in the absence of IL-33 led to reduced MRGPRX2 expression. As a result of its relative novelty, the network of signaling intermediates and transcription factors enabling MRGPRX2 expression exclusively in MC_TC_-type MCs is currently unresolved. Its partial reliance on JNK can set the stage for mechanistic studies into its regulation.

Paradoxically, IL-33-mediated downregulation of MRGPRX2 seemed to partially require the same kinase, while the process was independent of p38 action. In fact, the negative effect of IL-33 on MRGPRX2 expression was partially reversed after perturbation of JNK, as JNK silencing and IL-33 had no additive effects in terms of MRGPRX2 downregulation, but even conversely, the JNK-KD and IL-33 combined caused a somewhat higher MRGPRX2 expression without reaching baseline levels. Our study therefore highlights that the same kinase (JNK) can adopt opposing roles in the presence versus absence of IL-33. Reasons behind this dichotomy are currently unknown but may be associated with the strength of the JNK signal. In fact, steady-state activity of JNK is below detection by western blot in skin-derived MCs (Appendix A), even though some phosphorylated JNK can be measured by flow-cytometry, as shown by us recently [24]. This low activity may be sufficient to maintain functionality of transcription factors that drive MRGPRX2 expression (or alternatively, keep negative regulators at bay). IL-33 increases JNK phosphorylation, and this enhanced activity may now lead to the phosphorylation of a widened (and/or altered) set of effectors to mediate downregulation of the receptor. Interference with this latter aspect would partially reverse the effect of IL-33 without being able to restore MRGPRX2 expression to steady-state levels, because of the still-lacking baseline function of JNK. This is exactly what we have found. In this scenario, MCs may tune their JNK pathway to make it perform distinct functions in different contexts. Dependence of the role of JNK upon the cellular context has been reported, especially with regard to apoptosis versus survival decisions, as JNK has been associated with both of these opposing processes [56].

The negative regulation of the pseudo-allergic/neurogenic route by long-term IL-33 is notable, because SCF by itself dampens MRGPRX2 responsiveness and expression [16,19], so that on extended exposure, IL-33 further augments the negative effect of SCF and does not reverse the positive effect of SCF (as detected for the allergic route) [19,24,33].

The combined findings from our previous report [24] and this study indicate that long-term IL-33 may yield effects reminiscent of long-term exposure to glucocorticoids, as reported recently by Yamada and colleagues for murine CTMC [57]. In fact, we found reduced chymase mRNA, coupled with increased histidine decarboxylase/histamine abundance [24], and a nearly abrogated MRGPRX2 pathway in this report, a triad of manifestations likewise detected for dexamethasone [57]. IL-33 and glucocorticoids may thus employ overlapping strategies to cause this phenotypic switch in MCs.

We finally explored whether IL-33 impacts MRGPRX2-elicited exocytosis on short-term priming and found that this was indeed the case. By direct comparison with FcεRI-triggered release, which was similarly enhanced, we also disclosed what is, to our knowledge, the first mediator to uniformly affect allergic and pseudo-allergic/neurogenic degranulation pathways.

In fact, potent MC-modulating factors thus far studied (SCF, IL-4, and retinoic acid) have shown a clear-cut separation with opposite patterns of regulation, i.e., enhancing effects on the allergic route, but concomitant suppression of the MRGPRX2 pathway [16,19,20]. IL-33 is thus the first cytokine to support the MRGPRX2 route altogether.

On the other hand, priming effects of IL-33 on granule exocytosis, as such, have been reported by different laboratories and for distinct types of cells. While most research found no effect on degranulation of IL-33 alone, Kaur et al. showed that IL-33 could slightly degranulate MCs on its own, but the extent was modest [58]. Conversely, more groups reported on increased allergic stimulability in different types of MCs [25,26,27,28]. Increased degranulation by FcεRI-aggregation was likewise reported for basophils [59,60]. In the current study, we confirmed that allergic degranulation is supported by IL-33 also in skin-derived MCs (Figure 4).

In contrast to FcεRI-triggered degranulation, very little is known about if and how MRGPRX2-driven responses may be regulated by IL-33. To our knowledge, only Cop et al. studied MRGPRX2-triggered degranulation under the control of several cytokines, including IL-33, in CD34^+^-derived MCs; they found no effect on the pseudo-allergic/neurogenic route but confirmed a supportive effect on allergic degranulation [25]. It remains unclear whether the different outcome compared to our findings is due to the different signal strength mediated by MRGPRX2 in different types of MCs or, more likely, whether it is the result of distinct signaling machineries contracted by different MC subsets, including the degree of p38 activation. In skin MCs, as typical representatives of the MC_TC_-type MC and prominent producers of MRGPRX2, the positive effect of IL-33 on MRGPRX2 function could be consistently detected for both ligands and (nearly) all MC preparations employed.

Together with previous studies, which reported priming effects of IL-33 on alternative MC activation routes, including adenosine, C5a, and IgG [59,61], our present demonstration of strengthened MRGPRX2 function implies that the mechanisms by which IL-33 increases acute MC responses may be quite universal across degranulation-competent receptors beyond FcεRI.

Having found that receptor expression was unperturbed 30 min after IL-33 exposure, we assumed that IL-33 signaling engaged in a positive cross-talk with signals transduced by MRGPRX2 (as well as FcεRI). Examining whether JNK, p38, or both foster degranulation, we found that nearly the entire priming effect of IL-33 depended on p38, and was abolished by SB203580 (p38 inhibitor), whereas JNK played no apparent role. IL-33 is highly effective at p38 activation, and signal intensity may be key to the strengthened degranulability of MCs. A further hint comes from the difference with SCF, which dampens the MRGPRX2-driven route [16]. In fact, distinct signaling events are elicited by IL-33/ST2 vis-à-vis SCF/KIT in skin-derived MCs, as described recently: p38 activation uniquely occurs upon IL-33 stimulation and is undetectable after SCF triggering [24]. In contrast, ERK1/2 and AKT activation are limited to SCF and are below detection after IL-33.

The involvement of p38 is intriguing, because even though it forms part of the signaling cascade elicited by FcεRI-aggregation, p38 has been more commonly associated with cytokine production (especially TNF-α), lipid mediator generation, migration, proliferation, chemotaxis, and adhesion, but not degranulation in the first place [62,63,64,65,66,67,68,69,70]. We confirmed a lacking effect of p38 on degranulation elicited by the two routes in the absence of IL-33 (Appendix A), so p38 was only involved in the potentiation by IL-33. This argues that p38 activation by IL-33 is qualitatively or quantitatively different from the one triggered by FcεRI aggregation [71], or alternatively, that its activity needs to combine with other so far undiscovered events specifically elicited by IL-33, but not by FcεRI (or MRGPRX2). Taken together, this part of the study illuminated a novel regulatory role of the IL-33/p38 axis, which positively affected granule exocytosis during acute pseudo-allergic/neurogenic and allergic responses alike.

Our combined results support the concept of a clear segregation between long- and short-term effects exerted by IL-33. We recently reported that allergic skin MC degranulation is curbed when IL-33 is chronically administered [24], while we found here that IL-33 has a priming effect on FcεRI-aggregation when given acutely (Figure 4). We now report that the opposing roles of IL-33 apply even more so to MRGPRX2-triggered activation, which is virtually abolished in a chronic setting, yet supported when administered shortly before stimulation.

Our data reinforce the concept that MCs develop protection from overt and persistent IL-33 by shutting down degranulation to avoid exaggerated responses and restore tissue homeostasis under conditions of chronic inflammation. Conversely, an acute burst of IL-33 supports secretion, likely to foster MC-dependent defense strategies. An anti-inflammatory character on prolonged exposure to IL-33 is also highlighted by its resemblance with MC responses to sustained glucocorticoids [57].

The significance of the MC-IL-33 connection and necessity of MC protection against IL-33 hyperactivity is emphasized by the potent degradation of IL-33 by MC chymase released from skin MCs during degranulation, together with histamine [72,73]. In fact, IL-33 is one of the fewer (among a panel of) cytokines targeted for degradation by granule-associated proteases, suggesting a mechanism of negative feedback control [72]. The importance of chymase-mediated degradation of IL-33 was also found in vivo in an allergic airway inflammation model [74].

Along the same lines, a protective, anti-inflammatory function of the IL-33/ST2/MC axis has been described by different studies, including limitation of inflammatory monocyte responses [61], enhancement of regulatory T cells through enhanced IL-2 production [75,76], protection against airway hyperresponsiveness owing to PGE2 generation [77], and MC-dependent restoration of mucosal healing [78].

Collectively, IL-33 functions as a double-edged cytokine in skin MCs, which highlights its involvement in the tunable nature of MCs as mediators or suppressors of immune and inflammatory responses [79].

## Figures and Tables

**Figure 1 cells-08-00341-f001:**
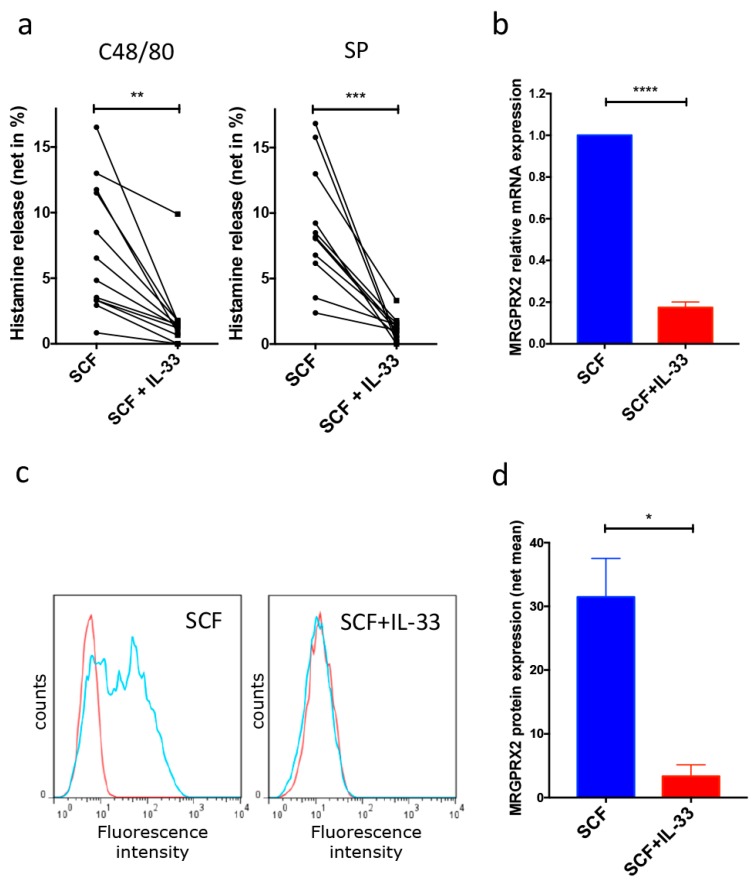
Chronic exposure to IL-33 abrogates MAS-related G protein-coupled receptor-X2 (MRGPRX2)-triggered degranulation through reduced receptor expression. Cells were cultured in SCF only or SCF and IL-33 for five weeks. (**a**) Net histamine release elicited by C48/80 and SP (*n* = 11). (**b**) MRGPRX2 relative mRNA expression (mean ± SEM, *n* = 15). (**c**,**d**) MRGPRX2 cell surface expression determined by flow-cytometry. (**c**) Representative histograms, red: Isotype, blue: MRGPRX2-specific antibody. (**d**) Cumulative data given as net mean fluorescence intensity (MFI) (MFI specific antibody − MFI isotype control) ± SEM of four independent experiments. * *p* < 0.05, ** *p* < 0.01, *** *p* < 0.001, **** *p* < 0.0001.

**Figure 2 cells-08-00341-f002:**
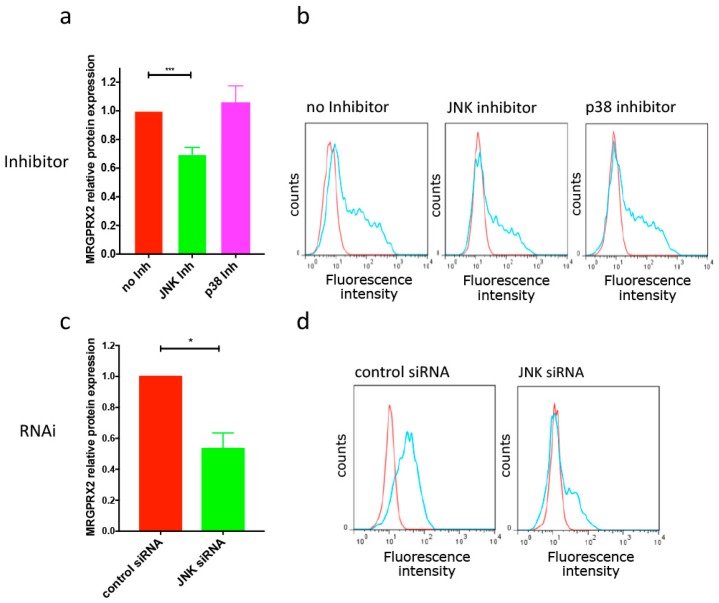
Interference with JNK attenuates MRGPRX2 expression in the absence of IL-33. (**a**,**b**) Cells were deprived of growth factors overnight, and treated with the inhibitor SB203580 (p38) or SP600125 (JNK) for 24 h. (**a**) Cumulative data from 16 independent experiments are given as net mean fluorescence intensity (MFI), normalized to control (no Inh). (**b**) Representative histograms, red: Isotype, blue: MRGPRX2-specific antibody. (**c**,**d**) Cells were subjected to RNAi targeting JNK1 by the Accell^®^ technology, a matching non-target siRNA served as control. (**c**) MRGPRX2 cell surface expression given as net mean fluorescence intensity, normalized to control (*n* = 5). (**d**) Representative histograms as above. * *p* < 0.05, *** *p* < 0.001. Inh = inhibitor.

**Figure 3 cells-08-00341-f003:**
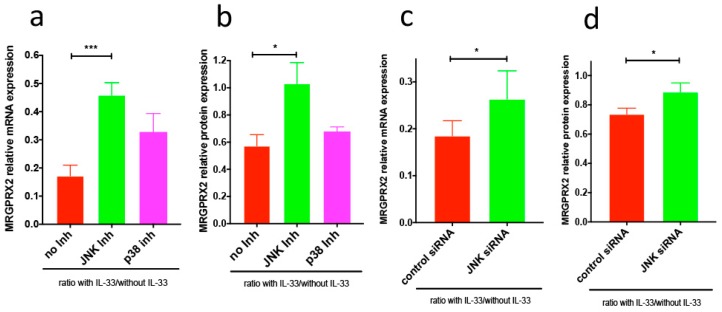
Perturbation of JNK function partially reverses the negative effect of IL-33 on MRGPRX2 expression. (**a**,**b**) After deprivation of growth factors overnight, cells were pretreated with the inhibitor SB203580 (p38) or SP600125 (JNK) for 15 min, followed by IL-33 stimulation. Cells treated with inhibitors alone served as controls. All data are presented as ratios “IL-33/no IL-33” (mean ± SEM) as in our recent publication [24]. (**a**) After 4 h, cells were harvested for MRGPRX2 mRNA, as determined by RT-qPCR (*n* = 7). (**b**) After 24 h, MRGPRX2 cell surface expression was determined by flow-cytometry (*n* = 16). (**c**,**d**) Cells pre-exposed to JNK-siRNA or control siRNA for 48 h were stimulated with IL-33 or left untreated. (**c**) RNA was taken after an additional 4 h, and MRGPRX2 mRNA expression quantified by RT-qPCR. (**d**) MRGPRX2 protein expression was measured after 24 h. For the effects of JNK-siRNA alone, see Figure 2. Data are the mean ± SEM of five to seven independent experiments. * *p* < 0.05, *** *p* < 0.001.

**Figure 4 cells-08-00341-f004:**
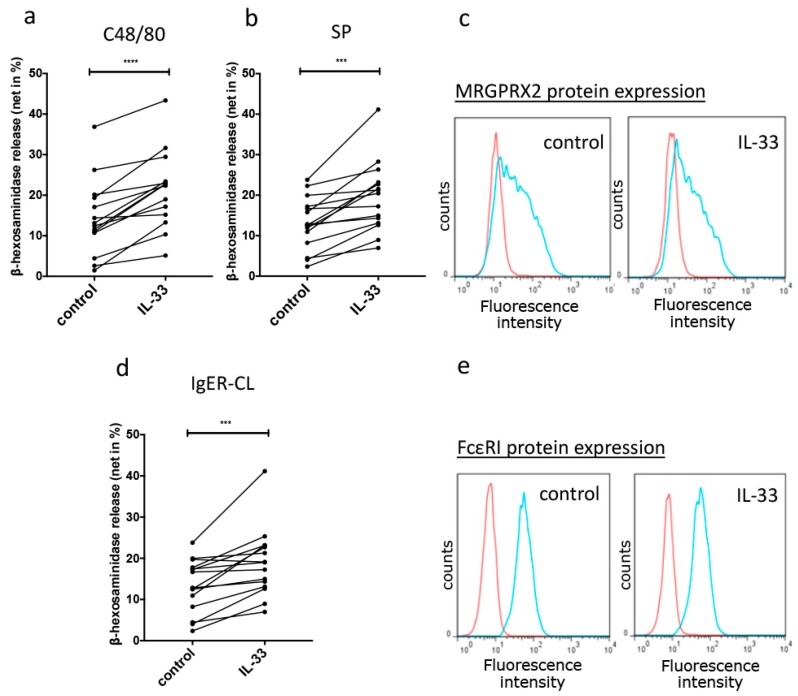
Short-term IL-33 primes pseudo-allergic/neurogenic and allergic mast cell (MC) degranulation. (**a**,**b**) Net β-hexosaminidase release activated by the MRGPRX2 agonists C48/80 (10 μg/mL) or SP (30 µmol/L) 30 min post-activation following 30 min priming with/out IL-33 in 14 independent experiments, each using a separate MC preparation. Dots representing the same preparation are interconnected. (**c**) MRGPRX2 protein expression at the cell surface after 30 min of IL-33 treatment. Representative histograms for control and IL-33. Red: Isotype control. Blue: Receptor-specific antibody. (**d**) Net β-hexosaminidase release activated by FcεRI-aggregation (AER-37, 0.1 μg/mL), given for comparison and primed with IL-33 as above. Data are the mean ± SEM of 14 independent experiments. (**e**) Representitive histograms showing FcεRI protein expression. *** *p* < 0.001, **** *p* < 0.0001.

**Figure 5 cells-08-00341-f005:**
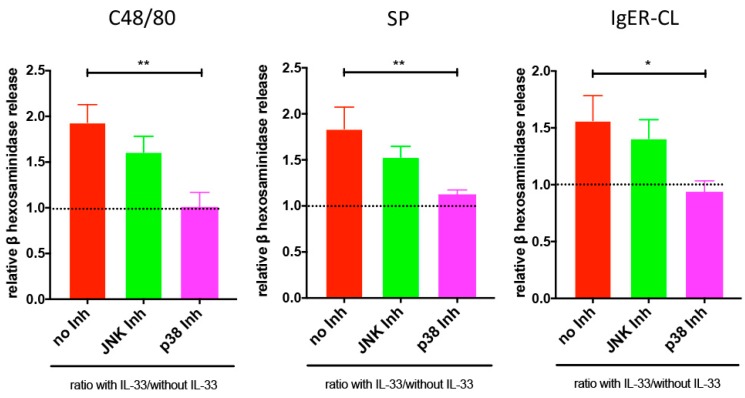
IL-33 priming occurs in a p38-dependent manner. Cells were pretreated with the inhibitors SB203580 (p38) or SP600125 (JNK) for 15 min, then with or without IL-33 for 30 min, and finally stimulated by C48/80 (10 μg/mL), SP (30 µmol/L), or FcεRI aggregation (AER-37, 0.1 μg/mL). The net release of β-hexosaminidase was determined (in %) and normalized to cells equally treated with inhibitors, but not exposed to IL-33. Cells treated with inhibitors alone served as controls (the results are given in Appendix A). The data are the mean ± SEM of seven independent experiments. * *p* < 0.05, ** *p* < 0.01.

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
