# Peer review of "IL-33 and MRGPRX2-Triggered Activation of Human Skin Mast Cells—Elimination of Receptor Expression on Chronic Exposure, but Reinforced Degranulation on Acute Priming"

_cells, 2019, doi:10.3390/cells8040341_

Round 1

Reviewer 1 Report

Dr. Babina’s group recently published a manuscript in the Journal of Experimental Dermatology showing that exposure of human skin mast cells to IL-33 for 5 weeks increases mast cell number but attenuates both FceRI expression and IgE-mediated mast cell degranulation. By contrast, IL-33 also induced histamine production and this effect is mediated via p38MAPK activation. In the present study, the authors show that acute treatment of human skin mast cells enhances MRGPRX2-mediated degranulation independently of receptor expression. By contrast, exposure of mast cells to IL-33 for 5 weeks eliminates MRGPRX2 expression, which is associated with substantial decrease in skin mast cell degranulation in response to substance P. The authors also examine the roles for JNK and p38MAPK signaling pathways on the acute and chronic IL-33 effects on MRPGRX2 expression and function.

Comments:

This study highly complements the group’s recent studies on IL-33/FceRI interaction. The data presented in the manuscript are convincing and likely to be important in the context of neurogenic inflammation, pain (Neuron101, 412-420, 2019) and chronic inflammatory skin diseases such as atopic dermatitis and idiopathic chronic urticarial.

Comments:

1.      The title of the manuscript implies that the authors have utilized drugs that induce pseudo-allergy. This is not the case. Since the focus of the study is on substance P/MRGPRX2 the title of the manuscript should be changed and the text of the manuscript should reflect substance P/MRPGRX2-mediated diseases rather than pseudo-allergy.

2.      Only one breast skin-derived mast cells were employed for studies in Fig.1a. This is a critical experiment and should be reproduced in other samples.

3.      IL-33 treatment for 4 hours reduces MRGPRX2 mRNA to maximum level. However, functional studies are performed with skin mast cells after 5 weeks of culture with IL-33. The reason for this difference needs to be explained or additional experiments performed.

Author Response

Reviewer 1

Comments and Suggestions for Authors

Dr. Babina’s group recently published a manuscript in the Journal of Experimental Dermatology showing that exposure of human skin mast cells to IL-33 for 5weeks increases mast cell number but attenuates both FceRI expression and IgE-mediated mast cell degranulation. By contrast, IL-33 also induced histamine production and this effect is mediated via p38MAPK activation. In the present study, the authors show that acute treatment of human skin mast cells enhances MRGPRX2-mediated degranulation independently of receptor expression. By contrast, exposure of mast cells to IL-33 for 5weeks eliminates MRGPRX2 expression, which is associated with substantial decrease in skin mast cell degranulation in response to substance P. The authors also examine the roles for JNK and p38MAPK signaling pathways on the acute and chronic IL-33 effects on MRPGRX2 expression and function.

Comments:

This study highly complements the group’s recent studies on IL-33/FceRI interaction. The data presented in the manuscript are convincing and likely to be important in the context of neurogenic inflammation, pain (‚) and chronic inflammatory skin diseases such as atopic dermatitis and idiopathic chronic urticarial.

We thank the reviewer for this summary and for finding merit in our study. We are grateful for their helpful suggestions for further improvement.

Comments:

The title of the manuscript implies that the authors have utilized drugs that induce pseudo-allergy. This is not the case. Since the focus of the study is on substance P/MRGPRX2the title of the manuscript should be changed and the text of the manuscript should reflect substance P/MRPGRX2-mediated diseases rather than pseudo-allergy.

We agree that we do not use drugs typically associated with pseudo-allergy like muscle relaxants or icatibant. However, we do not only use SP as an exogenous ligand but also C48/80 as the prototypical basic MC secretagogue and this is a typical exogenous ligand of MRGPRX2. Both pathways, i.e. the exogenous and the endogenous routes, are equally targeted by IL-33, so this is not a merely SP-centered study. However, we changed the title and do not refer to pseudo-allergy anymore, only to MRGPRX2.

IL-33 and MRGPRX2-triggered activation of human skin mast cells – elimination of receptor expression on chronic exposure, but reinforced degranulation on acute priming

With regard to the “neurogenic route”, we added this throughout the text and exchanged “pseudo-allergic” by “pseudo-allergic/neurogenic” in many spots or replaced pseudo-allergic by MRGPRX2-driven. Because the same receptor is at play, it is highly likely that (apart from nuances) regulation will be similar across ligand subgroups – this is also what we found. In particular, if MRGPRX2 expression is nearly eliminated (like after long-term IL-33), the route will be abolished altogether and cells not responsive anymore to c48/80, SP or to any other ligand (we actually have some further data on other ligands to be presented in future manuscripts – they are all consistent).

We have also extended the discussion to highlight MRGPRX2’s role in MC activation by neuropeptides. The interesting publication in Neuron (pointed out by the reviewer) was very helpful in this context and has been added to the reference list and is reffered to in the paragraph below.

“Because of the large and still increasing number of secretagogues acting by MRGPRX2 activation, the receptor has become the spotlight of MC research and is linked to a number of diseases triggered by non-immunological MC activation, including injection-site hypersensitivity reactions, chronic urticaria, atopic dermatitis, red man syndrome, periodontitis, and drug anaphylaxis [1-6]In particular, a bidirectional cross-talk between MCs and sensory neurons is widely appreciated, and MCs can be found adjacent to peripheral nerve endings, where they are involved in neurogenic inflammation. By acting as the receptor for a variety of neuropeptides (including Substance P, VIP [Vasoactive intestinal peptide], somatostatin, cortistatin and PACAP [pituitary adenylate cyclase-activating peptide] [3,4,7]), likely MRGPRX2 contributes to neurogenic inflammation. In fact, it was recently demonstrated that MRGPRX2 (and its murine equivalent) mediated mechanical and thermal hyperalgesia, as well as immune cell recruitment by MC cytokine/chemokine release triggered by SP [8]. This is in accordance with another report showing showing that intradermal injections of PACAP result in cytokine secretion promoting leukocyte recruitment in a MC dependent fashion [9].“

2.      Only one breast skin-derived mast cells were employed for studies in Fig.1a. This is a critical experiment and should be reproduced in other samples.

Perhaps there is a misunderstanding. In figure 1a, it was 4-15 independent experiments, not one single breast skin sample, as described in the fig. legend (copied below for convenience with “n” highlighted).

“Cells were cultured in SCF only or SCF + IL-33 for 5 weeks. a)Net histamine release elicited by C48/80 and SP (n=11). b)MRGPRX2 relative mRNA expression (mean ± SEM, n=15).c,d)MRGPRX2 cell surface expression determined by flow-cytometry. c)Representative histograms, red: isotype, blue: MRGPRX2-specific antibody.d)Cumulative data given as net MFI (MFI specific antibody - MFI isotype control) ± SEM of 4 independent experiments.”

If the reviewer aims at foreskin MCs as an additional type of skin MCs, we include data for 2 cultures that were started before submission of the manuscript, so that we could harvest them in the meantime (i.e. after 5 weeks). The data are presented as a new Figure S1.

The data perfectly match those of breast skin MCs (elimination of MRGPRX2 upon culture with IL-33), indicating that IL-33 driven downregulation is not restricted to one single skin location, sex or age group (in our laboratory, foreskin MCs are from juvenile subjects, while breast skins are from adult women undergoing breast reduction surgery).

In addition, the data given in Figure S2 and in main Figure 3 show downregulation of MRGPRX2 in foreskin MCs, even though at a much earlier time point.

Given the above, we do not believe that more samples are required to convince the reviewer about the existence of this dampening effect of IL-33 on MRGPRX2 also in foreskin MCs. However, if required, we are willing to perform more experiments here, but would need to extend the deadline by some 3 months in order to obtain sufficient skin samples and culture them for 5 weeks.

3.      IL-33 treatment for 4 hours reduces MRGPRX2 mRNA to maximum level. However, functional studies are performed with skin mast cells after 5weeks of culture with IL-33. The reason for this difference needs to be explained or additional experiments performed.

Thank you for pointing this out. In the first experiments, we used MCs cultured for 5 weeks to make the study comparable to our previous report published in JID, where we used a long-term model of chronic exposure to IL-33.

However, it would have been very difficult to follow-up on the mechanisms if the effect had truly required so much time to come into force.

Therefore, the idea was to run a kinetics series to identify the earliest time point at which robust downregulation is detectable. Fortunately, we found that MRGPRX2 expression goes down very rapidly at mRNA level, followed by the protein in a time-shifted way.

This was very helpful in terms of mechanistic studies, because, as mentioned above, it would have been problematic to culture cells with inhibitors for that long and even if this had been possible, interpretation of the data would have been extremely difficult due to various indirect effects occurring during the long period.

We now explain this better in the following paragraph:

 “We set out to address the mechanism beyond the notable downregulation of MRGPRX2. Because the use of inhibitors for prolonged times (like 5 weeks) would have been impractical, we first assessed with a time-course analysis after what time MRGPRX2 downregulation commenced following the addition of IL-33. This approach revealed that the decrease at transcript level was rapid (detectable at 2-4 h after the addition of IL-33) but still detectable at 48 h without re-addition of IL-33 (Figure S2)The 4 h-time point was selected for further experiments (and based on this, the 24 h point was chosen for the analysis of protein expression). The rapid response to IL-33 made pharmacological interference and knockdown experiments feasible without concerns about indirect effects (likely accumulating over a 5-week-period and precluding proper interpretation).”

Please note that we also have data from 2 cultures that were treated with SCF +/- IL-33 for 4 d (versus SCF alone). After this time, MRGPRX2 is strongly decreased (one example depicted below), even if not as completely eliminated as after 5 weeks (Figures 1, S1).

                        SCF (several weeks),                                     SCF (several weeks)

also last 4 d                                                     +then SCF+IL-33 (for 4 d)

We may summarize that MRGPRX2 expression goes down rapidly after contact with IL-33 but that downregulation is (nearly) complete after prolonged periods (5 weeks), a time at which basically no MRGPRX2 is detectable anymore (Figures 1, S1).

Reviewer 2 Report

The manuscript of Wang aims to elucidate the dual effect of IL-33 on MRGPRX2 expression. The authors showed that chronic exposure of human skin MCs to IL-33 reduces the expression of the receptor and consequently the MRGPRX2-dependent histamine release, whereas the acute exposure to the same amount of IL-33 enhances MRGPRX2-dependent as well as FceRI-dependent degranulation. The long and short-term IL-33 stimulation involves different members of the MAPK signalling pathway, JNK and p38 respectively.

The topic of the manuscript is appealing and is on line with a recent publication (Babina 2019 J invest Derm) of the same authors.  Experiments are well described and the work has the potential to make significant advance in the understanding of the pathways beyond allergic and pseudo-allergic degranulation of mast cells. 

I have only a major concern: I wonder how are BMMC after 5 weeks in IL-33 containing medium. Do they have the same granular content of BMMC cultured in absence of IL-33? Besides the release of histamine what is the impact of prolonged exposure to IL-33 on leukotrienes and/or cytokine secretion? Moreover, since 15 min of stimulation with IL-33 induce JNK and p38 activation (as shown in Fig S3) I wonder how are the levels of phosphorylation of p38 and JKN in BMMC cultured in presence of IL-33: are they similar to those of untreated BMMC? 

Personally, I would modify the order of the presented results, starting with short-term IL-33 experiments (partially already known) and concluding with long-term IL-33 experiments that represent the novelty of the work.

Minor concern

-figure 1C: FACS histograms lack axis titles

-at line 211: The adjective “intermediate” is not appropriate and densitometry analysis should be performed. 

-the blot images in figure S3 are not of good quality. Can the authors show another wb?

-Supplemental figure S2 and S3 must be inverted to match correctly with their mention in the text. Thus, wb must be shown in fig S2 and degranulation response in fig S3.

- Reference to supplemental figure S4 is missing. Actually it should be mentioned in the second paragraph and should be Fig S2. Please control the sequence of all the supplemental figures and remunerate them accordingly to their mentioning in the text .

Author Response

Reviewer 2

Comments and Suggestions for Authors

The manuscript of Wang aims to elucidate the dual effect of IL-33 on MRGPRX2 expression. The authors showed that chronic exposure of human skin MCs to IL-33 reduces the expression of the receptor and consequently the MRGPRX2-dependent histamine release, whereas the acute exposure to the same amount of IL-33 enhances MRGPRX2-dependent as well as FceRI-dependent degranulation. The long and short-term IL-33 stimulation involves different members of the MAPK signalling pathway, JNK andp38 respectively.

The topic of the manuscript is appealing and is on line with a recent publication (Babina 2019 J invest Derm) of the same authors.  Experiments are well described and the work has the potential to make significant advance in the understanding of the pathways beyond allergic and pseudo-allergic degranulation of mast cells.

We thank the reviewer for finding interest in the topic and for their very positive judgment.

I have only a major concern: I wonder how are BMMC after 5weeks in IL-33 containing medium. Do they have the same granular content of BMMC cultured in absence of IL-33? 

We assume that BMMC refers to breast-skin derived MCs and not to (murine) bone-marrow derived MCs (i.e. the way the abbreviation BMMC is more commonly used). The current study only uses human skin derived mast cells (from foreskin and breast skin tissue), no cells from the mouse.

Regarding granule mediators, they were actually one focus of our previous study (the one published in JID, Ref 24). We copy the relevant figure here for convenience (see below). While tryptase and chymase show some regulation at mRNA level in the presence of IL-33 (versus SCF alone), this does not seem translated to the mature, active peptidases. In contrast, histamine was positively regulated by IL-33, and so was its producing enzyme histidine decarboxylase. We therefore speculated in our previous study (Ref. 24) that the release of proteases versus histamine may be shifted after MCs have been exposed to IL-33.

We likewise insinuated that histamine released by FceRI-aggregation remains nearly stable in IL-33lowversus IL-33highenvironments due to IL-33’s opposite effect on IgER-triggered degranulation versus its effect on histamine abundance. We could demonstrate that this was indeed the case (Ref. 24).

However, FceRI-mediated secretion was reduced far less potently (and even increased in a few donors) in the previous study, while the MRGPRX2-triggered degranulation, shown in the current report, is almost eradicated by long-term IL-33.

Therefore, the higher histamine abundance will not be able to make up for the effect on degranulation and skin MCs exposed to IL-33 for longer periods will become almost refractory to MRGPRX2-triggered release of (all) preformed MC mediators, including histamine – if the in vitro findings mirror the in vivo situation, of course.

Please note this is the Figure 3 from our previous manuscript (Babina et al., JID2019, Ref 24), the legend is likewise copied here for convenience

Figure 3. Granule histamine contents is promoted by IL-33

Cells were cultured in SCF only or SCF+IL-33. Granule mediators were quantified at mRNA level as in Fig. 2 (mean ± SEM, n ≥ 16, upper panel) and at the level of the mediator itself (lower panel). a) MC tryptase transcript (upper); tryptase activity, as measured by cleavage of a specific substrate, n = 17 (lower). b) MC chymase transcript (upper); chymase activity, as measured by cleavage of a specific substrate, n = 14 (lower). c)histidine decarboxylase (HDC) transcript (upper) and cellular histamine content, n = 11 (lower). Dots representing the same donor are interconnected. ** p < 0.01, *** p < 0.001, **** p < 0.0001.

Besides the release of histamine what is the impact of prolonged exposure to IL-33 on leukotrienes and/or cytokine secretion? 

This is a very good point and we have actually started to follow up on these issues and expect to have some publishable results in 9-12 months.

Even though our lab has regular access to skin and therefore high numbers of skin MCs available, which we isolate 3-4 times per week, the cells are still limited, and we could not address all interesting questions at once.

We aimed at proliferation, survival, granule mediators, several phenotypical characteristics in the first manuscript in JID, as well as IgER-triggered degranulation. Now we focus on MRGPRX2 function comparatively after long-term chronic exposure to IL-33 vis-à-vis a short IL-33 pulse, and we clarify, at least in part, the underlying mechanisms.

With regard to cytokines, we have some preliminary data on IL-8, which is induced by short-term exposure to IL-33 alone (in accordance with data for other MC subsets published by other groups). We present this figure here to the reviewer. Note that these cells were cultured in SCF alone and THEN exposed to IL-33. We plan to also investigate whether MCs cultured in SCF+IL-33 behave differently from cells treated with SCF alone w.r.t. cytokine responses.

The aim of the experiments depicted below was to address the mechanism (i.e. whether IL-8 responses depend on JNK, p38 or both; it seems both). We will publish different aspects of IL-33 triggered cytokine responses collectively in one manuscript as soon as we have finished the experiments.

Note that IL-33 by itself induces cytokines (and this is what is depicted below). A different question will be how IL-33 impacts cytokine responses by IgER- versus MRGPRX2-triggering (which is also planned).

Fig. Left:

IL-8 is induced by IL-33 in skin mast cells, and the induction is JNK-dependent and potentially also p38-dependent

Cells were cultured in SCF only, deprived of SCF overnight and stimulated with IL-33 (20 ng/ml) for 2.5 h. Relative IL-8 mRNA expression is given as the mean +/- SEM of 7 independent experiments, normalized to non-stimulated control cells set as 1. One-way ANOVA with Dunnett's multiple comparisons test (all against non-target siRNA). ** p < 0.01. This is a preliminary figure. We are still working on the details.

Moreover, since 15 min of stimulation with IL-33 induce JNK and p38 activation (as shown in Fig S3)I wonder how are the levels of phosphorylation of p38 and JKN in BMMC cultured in presence of IL-33: are they similar to those of untreated BMMC?

Excellent point. We understand the reviewer asks whether, by some feedback mechanism, the cells become refractory to IL-33 treatment after long-term exposure (again, we believe the reviewer refers to human skin MCs).

Well, we had two long-term cultures running prior to submitting this manuscript, so it was possible to follow up on several of the points raised by the two reviewers within the 10-d-revision deadline.

We did not have enough cells left to run Western blots, though, but we could study this aspect by flow-cytometry in 2 cultures (pp38) and 1 culture (pJNK), respectively. We showed in the JID paper (Ref. 24) that both methods are comparable regarding IL-33 (and SCF) induced signaling in skin MCs. 

We found that after o/n growth factor deprivation, cells of both conditions still upregulate pp38 and pJNK 15 min upon IL-33, even though the MCs naive to prior IL-33 encounter seem to react more strongly (as would probably be expected). We disclose this information here to the reviewer (see fig. below).

To actually show it in the manuscript would require various repetitions (n of at least 8, probably 10) to find whether there is really a statistically significant difference between the two conditions.

If the reviewer finds this essential, we are certainly willing to put in the extra effort. However, to be able to run the experiments on a sufficient number of cultures, we would need some 3-4 months (considering skin availability and culture time).

For the time being, we may tentatively conclude that the cells do not become completely resistant to IL-33 stimulation, not even upon long-term contact to the cytokine, and this is also illustrated by the observation that MRGPRX2 downregulation is most pronounced after long time incubation with IL-33 (e.g. 5 weeks).

Personally, I would modify the order of the presented results, starting with short-term IL-33 experiments (partially already known) and concluding with long-term IL-33 experiments that represent the novelty of the work.

Thank you, this would have been an interesting option, and we actually thought about this. However, our idea was to match this manuscript to our recent publication [Ref 24]. There, we investigated the 5-week-culture period and assessed the IgER-mediated allergic route (among other aspects), so it seemed a “natural continuation” to start with the same conditions, but now focusing on the MRGPRX2-route.

In addition, we cannot find any indication that would support that the short-term effects are partially already known”. To the best of our knowledge there is only one single paper out on IL-33 and MRGPRX2-triggered release, and this one failed to find any impact [Cop et al., Ref 25]. We discuss the reasons for this in the paper and paste this paragraph here for convenience.

“In contrast to FceRI-triggered degranulation, very little is known about if and how MRGPRX2-driven responses may be regulated by IL-33. To our knowledge, only Cop et al. studied MRGPRX2 triggered degranulation under the control of several cytokines, including IL-33, in CD34+-derived MCs; they found no effect on the pseudo-allergic/neurogenic route but confirmed a supportive effect on allergic degranulation [25]. It remains unclear whether the different outcome compared to our findings is due to the different signal strength mediated by MRGPRX2 in different types of MCs or, more likely, whether it is the result of distinct signaling machineries contracted by different MC subsets, including the degree of p38 activation. In skin MCs, as typical representatives of the MCTC-type MC and prominent producers of MRGPRX2, the positive effect of IL-33 on MRGPRX2 function could be consistently detected for both ligands and (nearly) all MC preparations employed.”

And the one about p38-supported degranulation is given here:

“The involvement of p38 is intriguing, because even though it forms part of the signaling cascade elicited by FceRI-aggregation, p38 has been more commonly associated with cytokine production (especially TNF-a), lipid mediator generation, migration, proliferation, chemotaxis and adhesion, but not degranulation in the first place [62-70]. We confirmed a lacking effect of p38 on degranulation elicited by the two routes in the absence of IL-33 (Figure S5), so p38 was only involved in the potentiation by IL-33. This argues that p38 activation by IL-33 is qualitatively or quantitatively different from the one triggered by FceRI-aggregation[71], or alternatively, that its activity needs to combine with other so far undiscovered events specifically elicited by IL-33, but not by FceRI (or MRGPRX2). Taken together, this part of the study illuminated a novel regulatory role of the IL-33/p38 axis, which positively affected granule exocytosis during acute pseudo-allergic/neurogenic and allergic responses alike.”

Together, we believe that the short-term effect on the MRGPRX2-elicited route is a new finding, and especially that the dependence of this effect on p38 is of quite some interest –rather unexpected, too, considering the poor connection between p38 activity and granule exocytosis.

In order to keep the structure intact without introducing too many changes (we were only granted 10 d to accomplish this revision), we have kept the order, but are willing to make these edits if the reviewer and/or editor find an altered order more appropriate. 

Minor concern

-figure 1C: FACS histograms lack axis titles

Thank you for pointing this out. We added axis titles to all flow-cytometry histograms (fig. 1C. Fig. 4, suppl. Fig. 1).

-at line 211: The adjective “intermediate” is not appropriate and densitometry analysis should be performed.

*Densitometry has been performed and ratios are given directly in the fig (see also below).

*In addition, we added the following phrase to the Methods:

Bands were quantified by densitometry with the software ImageJ (National Institutes of Health, Bethesda, MD, USA) and the degree of phosphorylation was assessed by the following equation: Ratio of phosphorylated protein = densityphospo-protein / densitytotal protein.

*Even though in all of our blots we have consistently seen stronger phosphorylation signals for p38 than of JNK, we removed the words “intermediate” and “profound” from the phrase:

Here, we reproduced this pattern by demonstrating JNK and p38 phosphorylation 15 min upon IL-33 administration (Figure S3).”

-the blotimages in figure S3 are not of good quality. Can the authors show another wb?

We exchanged the blots for other ones. We copy the replaced fig here for convenience. The ratios phospho/total by densitometry are given below the blot images.

The figure legend has also been updated:

“JNK and p38 are phosphorylated upon IL-33 administration.MCs were deprived of growth factors overnight, then stimulated with IL-33 (20 ng/ml) for 15 min. Representative immunoblots for p-JNK, p-p38 are depicted and reproduce findings recently published by us [18]. Blots were stripped, and re-probed with antibodies against total JNK, and p38. The bands were quantified by densitometry and the ratios of phosphorylated/total protein are given in the figure.”

-Supplemental figure S2 and S3 must be inverted to match correctly with their mention in the text. Thus, wb must be shown in fig S2 and degranulation response in fig S3.

- Reference to supplemental figure S4 is missing. Actually it should be mentioned in the second paragraph and should be Fig S2. Please control the sequence of all the supplemental figures and remuneratet hem accordingly to their mentioning in the text .

We are sorry for the confusion created by not numbering the suppl. figures correctly. We changed their order (also considering the inclusion of a new Fig. S1) and mention them all in the text in the order of first appearance, including JNK-KD efficiency (which is now really Figure S4). We hope that we have eliminated all mistakes related to order and numbering of suppl. figures. 

Round 2

Reviewer 2 Report

The authors replied to all my queries providing exhaustive and detailed answers. I really appreciated their accuracy.